# Fractionation of Lycopodiaceae Alkaloids and Evaluation of Their Anticholinesterase and Cytotoxic Activities

**DOI:** 10.3390/molecules26216379

**Published:** 2021-10-22

**Authors:** Aleksandra Dymek, Jarosław Widelski, Krzysztof Kamil Wojtanowski, Vladyslav Vivcharenko, Agata Przekora, Tomasz Mroczek

**Affiliations:** 1Independent Laboratory of Chemistry of Natural Products, Medical University of Lublin, 1 Chodzki St., 20-093 Lublin, Poland; jwidelski@pharmacognosy.org (J.W.); krzysztofkamilw@gmail.com (K.K.W.); tmroczek@pharmacognosy.org (T.M.); 2Independent Unit of Tissue Engineering and Regenerative Medicine, Medical University of Lublin, 1 Chodzki St., 20-093 Lublin, Poland; vlad.vivcharenko@gmail.com (V.V.); agata.przekora-kusmierz@umlub.pl (A.P.)

**Keywords:** *Lycopodium* *clavatum* L., *Lycopodium annotinum* L., *Huperzia* *selago* L., *Lycopodiaceae*, AChE inhibitors, alkaloids, TLC bioautography, VLC, HPLC/ESI-QTOF-MS

## Abstract

In view of the abundant evidence that Lycopodiaceae alkaloids, including the well-known huperzine A (HupA), are among the potent acetylcholinesterase (AChE) inhibitors, an attempt was made to search for new compounds responsible for this property. For this purpose, three plant species belonging to the Lycopodiaceae family, commonly found in the Euro-Asia region, were subjected to the isolation of bioactive compounds, their identification and subsequent evaluation of their anticholinesterase and cytotoxic activities. Methanolic extracts of two *Lycopodium* and one *Hupezia* species were obtained via optimized pressurized liquid extraction (PLE) and then pre-purified using innovative gradient vacuum liquid chromatography (gVLC). For the first time, three sorbents of different porosity packed in polypropylene cartridges and mobile phase systems of different polarity were used to elute the target compounds. This technique proved to be a rapid tool for the obtainment of alkaloid fractions and allowed one to select the appropriate process conditions to yield potent AChE inhibitors in each of the species studied. More than 100 collected fractions were analyzed via HPLC/ESI-QTOF-MS, which enabled one to detect more than 50 compounds, including several new ones previously unreported. Some of them were present in high purity fractions (60–90% of the established purity). TLC bioautography assays proved that the analyzed species are rich sources of AChE inhibitors, but *H. selago* showed the highest anti-AChE activity. Additionally, the modified silanized silica gel sorbent used allowed one to isolate *L. clavatum* alkaloids more efficiently using an aqueous reversed-phase solvent system. Furthermore, the tested extracts from the three plant extracts were found to be safe, as they did not exhibit cytotoxicity to skin fibroblasts.

## 1. Introduction

Plant species from the club moss genus *Lycopodium* and *Huperzia* (Lycopodiaceae) are known to be a rich source of compounds belonging to the alkaloid group, possessing unique heterocyclic skeletons usually containing C_15_N_2_, C_16_N, C_16_N_2_ and C_27_N_3_. The alkaloids are quinolizine, or pyridine and α-pyridone type, which have attracted the interest of researchers [1,2,3]. These compounds possess numerous properties such as anti-inflammatory, antioxidant, anticancer and antimicrobial activities, among others [4,5]. In addition, they are known for important bioactivities such as the ability to inhibit acetylcholinesterase (AChE), an enzyme responsible for decreasing the concentration of a neurotransmitter acetylcholine (ACh) at the synaptic site in the brain [3,6,7]. This property has been used in attempts to treat neurological disorders characterized by memory impairment, cognitive dysfunction, behavioral disturbances and deficits in activities of daily living [8,9]. Pharmacological studies conducted since the 1980s have shown that the compound responsible for this therapeutic effect is huperzine A (HupA), which is a potent, reversible, and selective AChE inhibitor. Since the discovery of this alkaloid and its first isolation from *Huperzia serrata* (Thunb. ex Murray) Trevis, it has become a promising drug for the treatment of Alzheimer′s disease symptoms [10]. To date, numerous research groups are working to isolate this compound as well as new, potent alkaloids with novel structures and new mechanisms of actions from this species and from related plants [6,11,12,13]. 

A large part of the pioneering work on alkaloids of the Lycopodiaceae family was carried out in Canada in the middle of the 20th century [14,15,16]. This led to the isolation and identification of more than 200 alkaloids from 54 *Lycopodium* species, which, in the 1990s, were classified by Canadian scientists Ayer and co-leaders [17] into four main classes: lycopodine, lycodine, fawcettimine and miscellaneous alkaloids with the key compound phlegmarine (Figure 1) [2,3,18]. The compounds that belong to the lycopodine class and have a single nitrogen atom in their structure are the largest group, with more than 100 compounds. However, most of the isolated *Lycopodium* alkaloids with remarkably potent AChE inhibition belong to the lycodine class, for example, huperzines A and B, and *N*-methyl huperzine B, which have two nitrogen atoms in the skeleton. This is related to their unique structure and the fact that they fit well into the gorge of the AChE enzyme active site [2,19,20]. Therefore, the discovery of new and more lycodine-type alkaloids seems promising for the treatment of neurodegenerative diseases. 

Due to the great demand of modern medicine for a drug of natural origin, effective in both the symptomatic and causal treatment of neurodegenerative diseases, the search for potent AChE inhibitors has been the focus of our interest. In our laboratory, studies have been conducted to isolate these compounds from *Narcissus* species belonging to the Amaryllidaceae family, obtaining high recoveries of alkaloids such as sanguinine or well-known galanthamine [21,22,23]. The above discovery inspired us to continue our investigations involving the search for potent AChE inhibitors. The literature data clearly indicate the strong potential of these compounds in Lycopodiaceae plant materials. Many studies show that HupA, compared to current pharmaceutical AChE inhibitors on the market, including galanthamine, penetrates the blood–brain barrier better, has higher bioavailability after oral administration and is less toxic [24]. The discovery of the therapeutic properties of HupA, particularly as a drug candidate for the treatment of neurodegenerative diseases, has led to increased interest in these compounds. Currently, HupA is isolated from *Huperzia serrata*, occurring commonly in Asia. However, the uncontrolled harvesting of this species has reduced its original range and depleted its natural resources [10,25]. Therefore, the aim of our work was to search for alternative sources of HupA and new potent AChE inhibitors from three species commonly encountered in the Euro-Asia region, belonging to the Lycopodiaceae family: *Lycopodium clavatum* L., *Lycopodium annotinum* L. and *Huperzia selago* (L.) Bernh. ex Schrank et Mart (=*Lycopodium selago* L.). 

To date, there are few reports on the chemical constituents found in these species, and these works are mainly from the early 1950s [14,15]. A preliminary study to optimize the extraction process of two *Lycopodium* species was conducted in our laboratory. The novel pressurized liquid extraction (PLE) used proved to be a rapid, reproducible and very efficient method for alkaloid extraction [23]. No previous studies carried out on plant materials of the *Lycopodium* species have achieved such high recoveries of desired compounds [26]. The percentage of major alkaloids such as lycopodine (in *L. clavatum* extract) or annotinine (in *L. annotinum* extract) exceeded even 40%. Such satisfactory results generated the need for a method to further purify crude extracts and obtain fractions with individual bioactive components, which would allow their efficient isolation, bioactivity assays and subsequent structural determination of the target compounds. So far, the isolation and purification of Lycopodiaceae alkaloids from extract has been carried out using traditional methods such as column chromatography and preparative high-performance liquid chromatography (preparative HPLC) [27,28,29,30,31]. Sometimes, the crude alkaloid extract was subjected to multistep column chromatography procedures on media (Sephadex LH-20, aluminium oxide (Al_2_O_3_)), and then on silica gel. Elution, on the other hand, was carried out using huge amounts of solvent mixtures. These techniques proved to be tedious and time-consuming processes resulting in many fractions and then subsequent subfractions of alkaloids [27,32,33,34]. Using this method, Wang and co-authors [33,35] successively isolated alkaloids from *Lycopodium japonicum* Thunb. species, which allowed one to isolate new compounds such as lycojaponicumins A–E and nine new lycopodine-type alkaloids in the obtained fractions, but in trace amounts. In turn, Zhang and co-workers [36] developed a rapid and inexpensive method to separate and purify HupA and HupB from *Huperzia serrata*. The low polar macroporous resin SP850 was selected from eight types of resins during pre-purification. HupA and HupB were separated on a C18 column via preparative HPLC, and the recovery of these alkaloids was as high as more than 80%. Halldorsdottir and co-workers [19] performed purification of the extract of the aerial parts of the Icelandic *L. annotinum* ssp. *alpestre* via vacuum liquid chromatography (VLC) followed by chromatography using open silica gel columns and solid phase extraction (SPE) on silica gel with ammonia-containing eluents. According to these methods, ten alkaloids were isolated, including a previously unknown *N*-oxide of annotine. The isolated alkaloids were then evaluated for their in vitro AChE inhibitory activity. Only a few methods have been used so far, including the colorimetric method in accordance with Ellman [11,31,37] and the bioautographic TLC assays according to Marston [38,39]. Different, well-known AChE inhibitors such as tacrine [33,35], hupA [39,40,41] donepezil [42] or galanthamine [43] were used as positive controls. Nevertheless, only a few of the isolated alkaloids possessed the ability to inhibit AChE. The previously mentioned isolation of alkaloids from the species *L. japonicum* led to the isolation of five new fawcettimine-related alkaloids, but these lacked anti-AChE activity [35]. Similar results were obtained for pure lycopodine-type alkaloids, whose inhibitory activity was also very low, with the exception of annotine *N*-oxide [19,31]. This example demonstrates that the presence of an additional the *N*-oxide function clearly affected the inhibitory activity. 

Therefore, the aim of this study was to thoroughly analyze the alkaloid compounds and search for potent AChE inhibitors from three well-known *Lycopodium* species. The isolation of alkaloids was carried out via optimized PLE and the extracts were purified via innovative VLC. Three different sorbent systems in different ratios and different mobile phase systems were used for the first time. This technique allowed one to rapidly compare the results from each experiment, for the three species, and to obtain individual compounds in each fraction. All these samples were thoroughly evaluated using a liquid chromatograph coupled with a mass spectrometer (ESI-QTOF-MS) and assessed for AChE inhibition activity. For the first time, a new TLC bioautography method according to Mroczek [44] with the use of a modern derivatization chamber was utilized. In addition, an in vitro cytotoxicity test against human skin fibroblasts was performed for each of the three plant extracts.

## 2. Results and Discussion

### 2.1. Optimization of Alkaloid Fractionation and Isolation 

Three plant species belonging to the Lycopodiaceae family: *L. clavatum*, *L. annotinum*, *H. selago*, were subjected to extraction using the PLE method. The conditions of this process were previously carefully selected through many trials and studies in our laboratory, and the results were published [26]. Based on these studies, it was found that the highest percentage of *Lycopodium* alkaloids were isolated under high pressure and temperature (100 bar, 80 °C) from methanolic extracts, so these conditions were used in this approach. Due to the fact that *Huperzia* species are endangered in some regions, the amount of plant materials was limited [10]. Extracts of 100 mL each were prepared: 4 × 5 g from *L. clavatum*, 3 × 5 g from *L. annotinum* and 2 × 5 g from *H. selago*, which were further processed. The evaporated and concentrated plant extracts were mixed with a small amount of celite, which was used as an auxiliary filter material. The samples prepared in this way were applied to cartridge columns filled with suitable sorbents in different proportions. For the fractionation and isolation of Lycopodiaceae alkaloids, sorbents were used, which are readily available and economical and have worked well for the isolation of alkaloids from the species *Narcissus triandrus* L. *c.v. ‘Hawera’* [23]. However, this method was modified in this study by abandoning the glass column, which was tedious to pack, required a large amount of sorbent and the analysis of which took a long time. Therefore, the polypropylene cartridges were filled with silica gel 60 F_254_ and basic aluminum oxide (alumina-Al_2_O_3_) (150MeSh) in ratios of 1:3 and 3:1, and a new sorbent silanized silica gel, 60 H, was used to completely fill the column. The packing of the sorbents in the columns as well as the entire fractionation was carried out under vacuum, which allowed one to use fine pore size sorbents and to increase the rate of obtaining fractions. Each elution step started the same way for each experiment. First, conditioning of the packed column was performed to activate the column before applying the sample for analysis. Then, the sample along with the celite was carefully applied to the column and eluted with a pre-selected mobile phase system. The rate at which each fraction was obtained depended on the type and ratio of the sorbents used, as well as the composition of the mobile phase utilized. Gradient elution was necessary due to the different polarities of the various alkaloids. Therefore, chloroform–methanol–25% aqueous ammonia solution with the proportions 95:5:0.2 (*v*/*v*/*v*) was used a solvent gradient system, which enabled the separation of alkaloids according to their polarity in a relatively short time. Hydrophobic compounds were eluted in the first fractions, while compounds with higher polarity were obtained using a methanol–chloroform solution (1:1, *v*/*v*) as eluent. Since more *L. clavatum* raw material was available, an additional experiment was performed for this species using a different, new reversed-phase system. This technique differed in that the stationary phase was less polar than the eluting solvent. Silanized silica gel was used as the stationary phase, and the eluent was a mixture of 60% aqueous methanol solution with the addition of two drops of 10% aqueous tartaric acid. The addition of the acid was intended to ionize the compounds, which prevented the desired components from binding to the stationary phase, thereby reducing peak tailing and improving chromatographic separation. The column was rinsed with two portions of this system, with different percentages of 60% and 85% methanol, and finally with pure methanol solution. By using a gradient of less polar organic solvent, compounds were separated due to their different hydrophobic/hydrophilic character. Thus, inversely to previous experiments, the most polar compounds were isolated in the first fractions. The obtained fractions were analyzed via TLC on silica gel plates and compared under UV light. Based on the TLC profile, identical fractions were combined. Fractions with red fluorescence that showed no alkaloids but confirmed the presence of chlorophyll under UV light were discarded. The obtained fractions (104 in total) for each plant species from each experiment are shown in Table 1.

In the next step of the study, the alkaloid compounds present in each fraction were identified via high performance liquid chromatography/electrospray-ionization-time-of-flight-mass spectrometry (HPLC/ESI-QTOF-MS), followed by TLC bioautography for AChE inhibition.

### 2.2. LC-MS Identification of the Isolated Compounds 

The purified alkaloid samples obtained from all experiments (104 in total) from the three plant species were analyzed using HPLC/ESI-QTOF-MS. This is a reliable tool for the efficient determination of unknown compounds present in plant extracts, as documented in many of our works [23,26]. This technique has numerous advantages such as high sensitivity, high mass accuracy (below 1 ppm) and high reproducibility of the analysis. It allowed one to separate a complex mixture of compounds with a wide range of molecular weights based on hydrophilic interaction. The compounds were investigated through ESI-QTOF-MS in the positive ion mode and by analyzing collision-induced dissociation (CID) MS/MS spectra. More than 50 alkaloids were identified based on their accurate masses, fragmentation characteristics and the collected literature data.

The first step towards the identification of *Lycopodium* alkaloid structures present in the three plant species studied was their detection and subsequent assignment to the appropriate *Lycopodium* alkaloid type. For this purpose, high-resolution MS and MS/MS spectra were used, which clearly indicated the differences between alkaloids belonging to the lycodine and lycopodine types, which were isolated in the highest percentage. Based on the knowledge that the amounts of nitrogen in the skeleton of the two types are different, the accurate measurement of the compound molecular mass allowed us to obtain the molecular formula and speculate which type of alkaloid it was. Unlike lycopodine-type alkaloids with a single nitrogen atom, compounds belonging to the lycodine type had two nitrogen atoms. Alkaloids with two nitrogen atoms can be easily distinguished because their *m*/*z* values of protonated molecules are an odd number. Moreover, in the MS/MS spectra, lycodine-type alkaloids showed the presence of the most abundant fragmentation ions above *m*/*z* 150 due to the loss of nitrogen and adjacent atoms. As a result, even mass product ions were formed from the odd mass protonated molecule [M + H]^+^ due to the loss of NH (15 Da), NH_3_ (17 Da), CH_3_N (29 Da), or C_3_H_9_N (59 Da), respectively. In contrast, in the lycopodine types, fragment ions arising from the protonated molecule [M + H]^+^ formed numerous product ions below *m*/*z* 150. The presence of peaks in the characteristic region at *m*/*z* 145, 105 and 84 is evident. Moreover, the MS/MS spectra often showed a neutral loss of C_4_H_8_ (56 Da) resulting from ring cleavage and the appearance of abundant fragment ions at *m*/*z* 174 or 172 corresponding to the structure of three connected six-membered rings. The presence of an additional hydroxyl group at this ring resulted in an intense peak at *m*/*z* 192 or 190. This is especially evident for MS/MS spectra of the alkaloids: acrifoline or lycopodine *N*-oxide. The latter alkaloid, with the protonated molecule at *m*/*z* 264, and molecular formula of C_16_H_25_NO_2_, also appears as a dimer at *m*/*z* 527. This fact is very important for the identification of *N*-oxide forms of new alkaloids, which often create dimeric forms. This knowledge allows one to preliminarily identify known and new lycodine- and lycopodine-type alkaloids from the general alkaloid group. Last but not least, CID MS/MS spectra are presented for the individual compounds detected in the analyzed fractions from each experiment for each species studied. In addition, fragmentation pathways of the protonated molecules were proposed to accurately analyze the qualitative composition of the compounds (Appendix A). Furthermore, the alkaloid content relative to the total amount of compounds in the individual fraction was calculated from the intensities of the peak areas detected on the base peak chromatograms (BPCs). These data are presented in Tables for each species, highlighting the VLC procedure used (Table 2, Table 3, Table 4, Table 5, Table 6, Table 7, Table 8, Table 9 and Table 10).

High alkaloid contents were identified in samples obtained from three experiments for *L. annotinum* species. In almost all fractions, the sum of isolated alkaloids was high and reached almost more than 90%. Moreover, in the second experiment, where the sorbent system was alumina and silica gel in a ratio of 3:1, in the first five fractions the dominant alkaloid was annotinine, accounting for 60–89% of the fraction (Table 3). In the next samples (fractions 6–9), this alkaloid was no longer detected, while lycopodine was obtained in large amounts (fraction 6, more than 70%). In the first experiment, where the sorbent ratio was reversed (alumina and silica gel in a ratio of 1:3), annotinine also dominated in the initial fractions (2 and 3) (Table 2). Annotine *N*-oxide also appeared, which was only detected in this experiment and was present in a high percentage in fraction 4 (more than 40%). In contrast, lycopodine and its *N*-oxide with high recovery (fractions 8–10) only appeared in the second half of the obtained fractions. This fact was related to the choice of a non-polar mobile phase and the increase in the polarity of the obtained compounds in the subsequent fractions. Such a high percentage of dominant alkaloids in particular fractions proved the appropriately chosen conditions and procedures of fractionation. In experiment 3, where silanized silica gel was used as a sorbent, similarly as in experiment 1, the lowest percentage of alkaloids was observed in the first fraction (Table 4). However, purifying and obtaining single alkaloid fractions was not as effective as in previous experiments. Yet, the compound *N*-methyl lycodine present in the last fractions of this experiment was detected for the first time. This discovery was satisfactory due to the fact that the introduction of *N*-methyl function, in contrast to their native derivatives, significantly affected the bioactivity of the compound. In addition, two new, previously unknown compounds were detected, and attempts will be made to identified them in future studies.

High percentages of total alkaloids in each sample were also obtained in two experiments conducted using *H. selago* extracts. The dominant compounds in these fractions were mainly derivatives of lucidine B (=serratanine) such as dehydrolucidine B, oxolucidine B or dehydrooxolucidine B. This was especially evident in experiment 1, where the group of these four alkaloids in samples 1–10 represented almost the entire fraction. These compounds, due to their structural diversity, belonged to the miscellaneous type of alkaloids. Their common molecular formula consisted of C_30_N_3_. The CID MS/MS spectra and the proposed fragmentation schemes of these compounds are shown in Appendix A). In subsequent fractions, the percentage of these alkaloids decreased due to the detection of alkaloids belonging to the lycopodine type such as selagoline or lycopodine *N*-oxide. On the other hand, in the second experiment, there was no such division, and selagoline and lycopodine *N*-oxide were present in almost all fractions, together with lucidine B derivatives, in a high percentage. In both experiments, new compounds of unknown structure, not yet described in the literature, were also detected and will be the subject of future studies. Moreover, alkaloids belonging to well-known and potent AChE inhibitors such as HupA and HupB and with moderate inhibition such as 16-hydroxyhuperzine B were identified. However, the amounts of these alkaloids were slight.

In the case of *L. clavatum* extracts, the VLC method yielded fewer final fractions compared to the previously mentioned species. This is due to the fact that on the basis of the TLC profile of the tested samples, some fractions were omitted owing to large amounts of ballast substances. This also resulted in a much lower percentage of identified alkaloids in each fraction. However, other compounds such as lycoclavanin or 16-oxolyclanitin belonging to the triterpenoid group were present. There were no dominant alkaloids in the individual fractions, and even compounds commonly detected in this species were present in small amounts. Alkaloids such as lycopodine or lycopodine *N*-oxide did not exceed levels of more than 10% in individual samples. However, new compounds were isolated, with novel structures that were not previously known. An example was a compound at *m*/*z* 679.5133, which was present in every fraction in the first three experiments and which was previously mentioned as an unidentified compound by Tian and co-authors [4]. 

An additional fourth experiment was performed for this species using reversed mobile phase systems with a higher polarity than before. The sum of identified alkaloids for this species was the highest in this case. In addition, purification proved to be the most effective in this experiment, as no compounds other than the desired alkaloids were identified. Moreover, fractions with dominant alkaloids were obtained. In the second fraction, the compound 8β-hydroxylycoposerramine had the highest percentage (above 46%). Moreover, a high percentage of more than 30% was calculated for lycopodine in most fractions. Comparing this experiment with experiment 3, in which the same stationary phase system (silanized silica gel) was used, and the distinguishing feature was the mobile phase system, a different quantitative and qualitative composition of the compounds was obtained. In experiment 3, the amounts of alkaloids detected were slight. It is interesting to note that the percentage of all detected compounds decreased with each fraction. The opposite is true in experiment 4. This is related to the polarity of the compounds and the fact that non-polar compounds are more readily isolated. The modified silica sorbent made better use of aqueous solvent systems to separate extremely non-polar substances and analyze specific polar compounds.

### 2.3. Anticholinesterase Assay

The AChE inhibiton assay used in the study was originally developed by Marston [38], but Mroczek [44] modified it to improve the sensitivity of the method. The use of 2-naphtyl acetate as the enzyme substrate proposed by Mroczek allowed one to obtain more concise and stable zones of inhibition on TLC plates. Scientists successfully applied this improved method in our laboratory to confirm the anti-AChE activity of alkaloid fractions obtained via VLC from *Narcissus* species [23]. Based on numerous literature data describing the potent AChE inhibitory activity present in the studied plant species of the Lycopodiaceae family, a rapid bioautographic method was used to screen and confirm the presence of the active compounds. In this method, 2-naphtyl acetate was added to the mobile phase composed of chloroform–methanol–25% aqueous ammonia solution at a ratio of 95:5:0.2 (*v*/*v*/*v*), developing TLC chromatograms. The substrate was converted into 2-naphtol by the enzyme and then a deep purple background of the TLC plates was obtained by reacting 2-naphtol with Fast Blue Salt solution. As a result, active AChE inhibitors were visible as colorless or yellowish zones on a purple background. The results are shown in Figure 2, Figure 3 and Figure 4. These analyses were performed in a modern and automated device, the Camag Derivatizer, which significantly exceeded the efficiency, speed and convenience of manual spraying. The reduction in the total analysis time allowed one to perform a series of bioautographic enzyme assays in a short time and compare the white zones of AChE inhibitors present in individual fractions obtained from each experiment. 

When tentatively analyzing the TLC bioautography results, by far the strongest zones of AChE inhibition are seen on TLC plates made for fractions obtained from *H. selago*. These data are also confirmed by the literature, where it is repeatedly mentioned that this plant species of the Lycopodiaceae family possesses strong AChE inhibitors, and it is the main source of HupA [25]. The results are shown in Figure 2. All fractions obtained from the two experiments show potent areas of AChE inhibition. Interestingly, it was observed that a strongly inhibitory alkaloid migrated at the same R_f_ (approximately in the middle of the TLC plates) for all fractions in the two *H. selago* experiments. Furthermore, it is readily apparent that the initial fractions contain inhibitors above this zone, while in the final fractions (12 and above), inhibitors are also present below this zone. On the basis of LC-MS data, it can be tentatively assumed that the white, intense zones of inhibition are not at all indicative of the presence of well-known AChE inhibitors such as HupA and its derivatives, because they were isolated in a small percentage and in a few fractions. In the first experiment, HupA was isolated in fractions 11–15, and in the second, it was present in fractions 12–14. In contrast, HupB and its derivative were only detected in the first experiment. These compounds, in contrast to HupA, are considered to be alkaloids with moderate inhibitory activity [24]. Nevertheless, strong zones of inhibition are seen in all fractions, and moreover indicate the presence of several inhibitors. Ortega and co-authors [20] came to similar conclusions by analyzing an extract obtained from the species *Huperzia saurus* (Lam). Trevis. In vivo studies confirmed its activity against AChE inhibition, but GC-MS analysis of the extract was surprising because no known potent Lycopodiaceae inhibitors were isolated. Thus, *H. selago*, like *H. saurus*, appears as a species with novel compounds with clear anti-AChE activity. It can be assumed that the compounds responsible for this activity could be: selagoline, lycopodine *N*-oxide or lucidine B derivatives, which are present in high percentages in each of the tests carried out on *H. selago* (Section 2.2). Moreover, in both experiments, the predominant lucidine B derivative compounds were recorded in the first pre-purified fractions. These samples also exhibited anti-AChE activity in TLC bioautographic assays. The presence of these alkaloids was known for a long time even in the species *Lycopodium lucidulum*, but their structures were unknown, and it was difficult for scientists to determine them. Moreover, there is not much information on any attempts to study AChE inhibitory activity. According to Tori and co-authors [45,46], these compounds lack anti-AChE activity. However, there is still insufficient data. In turn, data from the literature report that compounds belonging to lycopodine-type alkaloids are poor AChE inhibitors compared to HupA and HupB and their analogues. Many years of modeling studies have led to the conclusion that lycopodine-type alkaloids fit well into the gorge of the active site of the AChE enzyme, but the position of their functional groups does not allow them to establish strong interactions in the form of hydrogen bonds [19]. However, researchers have shown that some of these compounds, as a result of introducing an additional function such as *N*-oxide, can significantly affect this activity [19,20]. 

Significantly weaker white zones of inhibition are found on TLC plates for samples obtained from two *Lycopodium* species. Fractions obtained via VLC for *L. annotinum* were subjected to the TLC bioautography approach (Figure 3). White zones on the purple background were observed on each TLC plate, confirming the presence of AChE inhibitors. Only in fractions 1, obtained from three experiments, were no compounds with anti-AChE activity observed. These results are consistent with those obtained via LC-MS analysis (Section 2.2), where the lowest percentage of alkaloids or no alkaloids were detected in fractions 1 (in experiments 1 and 3). Interestingly, in the second experiment, where the alkaloid annotinine was dominant in fraction 1 with a percentage as high as 85.2%, no inhibition was observed either (Figure 3—LAN-2). These results confirm the studies published by Halldorsdottir [19], based on which the only alkaloids with strong anti-AChE activity detected in *L. annotinum* species are anhydrolycodine and annotine *N*-oxide. This annotine derivative was only detected in fractions 3–7 and 14 of the first experiment. These results seem to be consistent with the inhibition present on TLC plates, where white zones appear at the same R_f_ in these fractions. Each of the experiments performed confirmed the presence of compounds with anti-AChE activity in the individual fractions. Interesting results were observed in the third experiment, where two zones of inhibition were visible in fractions 3–8, and single inhibitors were obtained in the following fractions (Figure 3—LAN-3).

For *L. clavatum* species, four TLC bioautography approaches were performed for the obtained fractions from each experiment (Figure 4). The first two ones performed for the samples obtained from the first and second experiments show that AChE inhibitors were present in each fraction. In addition to the white zones of inhibition, dark purple spots are noticeable, which also indicate the presence of compounds with anti-AChE activity. The activity of these compounds is slightly weaker compared to the compounds present in the previously discussed species. Analyzing the data obtained from the LC-MS analysis, this may be related to the relatively small amounts of individual compounds detected in each sample. Moreover, the third experiment conducted for the fractions obtained via VLC using silianized silica gel, 60 H, as a sorbent gave the weakest results. The observed zones of inhibition overlapped and what is more, they were only visible in the first and last fractions (Figure 4—LCL-3). Spectacular results were expected from the fourth experiment performed for the fractions obtained via VLC with the new reversed-phase system. In each fraction, active compounds were observed that migrated at an R_f_ not previously achieved for this species in any experiment performed (Figure 4—LCL-4). Unfortunately, it can also be observed that in the vast majority of fractions strong zones of inhibition are seen at the start. This is not related to overloading of the fractions, but rather it can be assumed that the compounds with the non-polar mobile phase did not migrate but remained at the spot. Accurate qualitative separation and visibility of individual bands was also observed in this experiment. This may indicate better purification of the fractions via VLC and separation of the alkaloids using the new reversed-phase mobile system. To date, only a few studies evaluating the anti-AChE activity of this species are known. However, to our knowledge, biological studies involving individual alkaloid fractions have not been reported in the literature so far, and studies have been conducted mainly on crude plant extracts of *L. clavatum*. According to Orhan [42], the triterpenoid α-onocerin obtained from *L. clavatum* extract was responsible for the AChE inhibitory effect in vitro. However, later studies did not confirm this, and proved that another triterpenoid, lyclavatol, was responsible for this activity [47]. Our study shows that this species is not devoid of anti-AChE activity and the compounds responsible for it, and that the search for AChE inhibitors in *Lycopodium* species is entirely justified.

### 2.4. Cytotoxicity Assay

The satisfactory results of AChE activity obtained for the alkaloid fractions encouraged us to test the safety of extracts from these plant materials. Cytotoxicity tests revealed that *H. selago* and *L. annotinum* extracts were non-toxic to skin fibroblasts at the tested concentration range (2–500 μg/mL). Interestingly, the highest concentrations of mentioned extracts positively affected cell metabolism and fibroblast viability was significantly higher compared to the control cells (Figure 5). *L. clavatum* extract was non-toxic to skin fibroblasts at low concentrations (in the range of 2–31.3 μg/mL). Moreover, concentrations in the range of 2–7.8 µg/mL significantly increased cell viability compared to the control fibroblasts. Beginning from the concentration equal to 62.5 μg/mL, cell viability was significantly reduced to approximately 45–55% compared to the control cells.

## 3. Materials and Methods

### 3.1. Plant Materials

Oven-dried, powdered plants of three species belonging to the Lycopodiaceae family: *Lycopodium clavatum L*., *Lycopodium annotinum L*. and *Huperzia selago L*., were used in this research. Whole plant specimens were collected in southern Poland and western Ukraine. The plant material of *L. annotinum* was of Polish origin and it was collected in the Tatra Mountains. The other two species were of Ukrainian origin from the Oblast Rivne: *L. clavatum* from the Sarny region and *H. selago* from the Volodymyrets region. The species of these plants were authenticated by the authors of this paper and deposited in the Independent Laboratory of Chemistry of Natural Products, Medical University of Lublin in Poland.

### 3.2. Plant Materials Extraction

After grinding the dried three Lycopodiaceae species, 5 g of powder from each plant material was weighed and extracted twice via the PLE method using a Dionex ASE 100 extractor (Sunnyvale, CA, USA) in three cycles, 10 min each. The samples were placed in a 34 mL stainless steel extraction cell and extracted at an elevated temperature of 80 °C and 100 bar pressure, using methanol as a solvent. This extraction procedure and conditions have been previously tested and optimized, and the results have been described in detail [26]. The extracts obtained were transferred into 100 mL volumetric flasks and diluted to the mark with the solvent used for extraction. As a result, 4 extracts of *L. clavatum*, 3 extracts of *L. annotinum* and 2 extracts of *H. selago* were obtained and subjected to purification and fractionation processes. One additional 50 mL extract was prepared for each species, which was evaporated and destined for further studies to evaluate their cytotoxicity.

### 3.3. Sample Preparation and Purificatoin by VLC

Sample preparation and extract fractionation procedures using VLC were optimized and described in detail in work on the identification of potent AChE inhibitors from *Narcissus c.v. ‘Hawera’* [23]. Based on this, the purification and isolation of Lycopodiaceae family alkaloid fractions was carried out using the same polar sorbents: silica gel 60 F_254_ and basic Al_2_O_3_ (150 MeSh) as two-state phase gradient systems in different proportions and in a different order. In addition, a new sorbent was used for comparison: silica gel 60 H silanized, with which the column was fully packed. A polypropylene cartridge (dimensions: height: 17.5 cm and diameter: 3 cm), was filled alternately with Al_2_O_3_ and silica gel with the volume ratio of 1:3, and in subsequent trials with the volume ratio 3:1. The columns filled in this way were used to fractionate the extracts of the three plant species from the Lycopodiaceae family. Due to a larger amount of collected raw materials of two *Lycopodium* species, extracts were additionally fractionated for them using a polypropylene cartridge filled entirely with silanized silica gel 60 H. The prepared columns were first conditioned with two 100 mL portions of a 98:2 (*v*/*v*) solution of chloroform–methanol to activate the column before applying the sample to be analyzed. All the previously prepared plant extracts were evaporated to dryness on a rotary vacuum evaporator at 50 °C and then accurately transferred to the column as a suspension with celite added. VLC was carried out in solvent gradient systems using a chloroform–methanol–25% aqueous ammonia solution in proportions 95:5:0.2 (*v*/*v*/*v*). Fractions of 10 mL were collected by rinsing the analyte with 100 mL of mobile phase followed by 100 mL of methanol–chloroform solution (1:1, *v*/*v*), and finally the column was rinsed with 100 mL of pure methanol solution. For *L. clavatum*, an additional experiment was performed, and a new reversed-phase system was used. The individual steps were carried out in a similar manner, except that the column was conditioned with a 60% aqueous methanol solution with the addition of two drops of 10% aqueous tartaric acid (about 40 µL) and the mobile phase was a system of two 100 mL solutions: 60% and 85% aqueous methanol solution with the addition of two drops of 10% aqueous tartaric acid. The column was then flushed with 50 mL of pure methanol solution to wash the column. Finally, for *H. selago* species, 2 experiments were carried out using two sorbents in different proportions and for *L. annotinum*, 3 experiments. For *L. clavatum*, on the other hand, 4 experiments were conducted using different sorbents in different proportions and using two different mobile phase systems. The obtained fractions were analyzed via TLC on silica gel plates and the identical fractions were collected together based of their TLC profile under UV light.

### 3.4. HPLC/ESI-QTOF-MS Analysis of the Obtained Fractions

High performance liquid chromatography/electrospray-ionization time-of-flight-mass spectrometry (HPLC/ESI-QTOF–MS) analysis was performed to assess the identity of the active constituents of the fractions (104 in total) obtained via VLC for three plant species of the Lycopodiaceae family using a mass spectrometer. The purified samples were analyzed with an Agilent Technologies 6530B system in positive ion mode with an ESI-Jet Stream^®^ ion source and an Atlantis HILIC analytical silica column (dp = 3.5 µm, 150 × 2.1 mm) (Waters Milford, MA, USA) thermostated at 25 °C was used in accordance with previously described methodology [26]. The mobile phase was composed of solvent A: acetonitrile with the addition of formic acid (0.2%) and solvent B: water with the addition of formic acid (0.2%). The gradient procedure was: 0–10 min, 100% using solvent A; 10–40 min, 92% solvent A and 8% solvent B; 40–45 min, 64% solvent A and 36% of solvent B, 10 min of post time with initial parameters of gradient. Total analysis time was 55 min with a stable flow rate at 0.25 mL/min. The injection volume was 1 µL of fractions. The following settings of the mass spectrometer were applied: dual spray jet stream ESI ion source in positive ion, fragmentor voltage 120 V, nitrogen flow 10 L/min, gas temperature 350 °C, sheath gas temperature 325 °C and sheath gas 10 L/min, and nebulizer pressure was set at a level of 35 psi. The range of measured *m*/*z* was 100–1000 units in Auto MS/MS acquisition mode. Skimmer voltage 65 V and octopole RF Peak 750 V. CID was conducted at different energies, 10, 30 and 40 eV, with an MS scan rate of 1 spectrum per s, 2 spectra per cycle. Mass Hunter Workstation 2.2.1 software (Agilent Technologies, Santa Clara, CA, USA) running under a Windows system was used to handle the data obtained.

### 3.5. TLC Bioautography towards AChE Inhibitors

The detection of AChE inhibitory activity of Lycopodiaceae alkaloids was carried out using the modified Fast Blue B Salt procedure. This method was described in detail for the detection of anti-AChE activity of Amaryllidaceae alkaloids by Mroczek [22,23,44]. Solutions of all fractions obtained from VLC were spot-applied onto TLC plates (0.2 thickness) covered with silica gel, which were developed in the mobile phase of 95:5:0.2 (*v*/*v*/*v*) mixture of chloroform–methanol–25% aqueous ammonia solution with the addition of naphthyl 2-acetate (30 mg per 20 mL mobile phase). The mobile phase was prepared in a separating funnel, mixed, and the lower organic phase was taken for the plate development. They were then dried at room temperature and sprayed in an automated device for TLC derivatization as follows: AChE (3 U/mL in Tris buffer pH 7.8 stabilized with bovine serum) was incubated for 15 min at 37 °C and then sprayed with Fast Blue B Saltwater solution (1.25 mg/mL). After about 1 min, white zones of inhibition could be easily detected on a deep purple background.

### 3.6. Cytotoxicity against Human Skin Fibroblasts

The in vitro cytotoxicity test was carried out using normal human skin fibroblasts (BJ cell line) obtained from American Type Culture Collection (ATCC-LGC Standards, London, UK). The cells were grown in EMEM medium (ATCC-LGC Standards, London, UK) supplemented with 10% fetal bovine serum (FBS, Pan-Biotech, Bavaria, Germany), penicillin (100 U/mL), and streptomycin (100 μg/mL) (Sigma-Aldrich Chemicals, Warsaw, Poland). Fibroblasts were maintained in standard culture conditions: 37 °C, 95% humidity, 5% CO_2_. The BJ cells were seeded into 96-multiwell plates at a concentration of 3 × 104 cells/well. After 24 h of incubation at 37 °C (when cells were well spread and attached to the bottom of the wells), the culture medium was discarded and replaced with 100 µL of different concentrations of tested plant extracts (*H. selago*, *L. annotinum* and *L. clavatum*). The stock solutions (50 mg/mL) of all extracts were prepared in ethanol. The highest tested concentration was equal to 500 µg/mL (prepared in culture medium). Other concentrations were obtained by eight serial 2-fold dilutions using culture medium. Fibroblasts maintained in the culture medium without plant extract served as a negative control of cytotoxicity. After the 24 h exposure of fibroblasts to tested plant extracts, their viability was determined using an MTT assay (Sigma-Aldrich Chemicals, Warsaw, Poland) as describe previously [48]. The experiment was repeated in 5 separate experiments (n = 5). Statistically different results (considered at *p* < 0.05) compared to the control cells were determined using an unpaired t-test (GraphPad Prism 8.0.0 Software).

## 4. Conclusions

In the present study, for the first time, an attempt was made to obtain pre-purified Lycopodiaceae alkaloid fractions combined with bioactivity testing and cytotoxicity evaluation of the extracts. To date, bioactivity studies have been mainly conducted on plant extracts [11,12,42,43]. Moreover, the techniques used have not been previously utilized to isolate and evaluate the activity of Lycopodiaceae compounds. The pre-purification and fractionation of the extracts was carried out using the innovative VLC method. This technique proved to be effective, and the preparation of polypropylene cartridges did not require too much time and effort. Three readily available, low-cost sorbents and mobile phase systems with different polarities were used. This resulted in more than 100 fractions, which were then subjected to identification and bioactivity studies. More than 50 Lycopodiaceae alkaloids were detected in the obtained samples using HPLC/ESI-QTOF-MS, including several new compounds not previously reported in the literature. Some of them were present in high purity fractions (60–90% of the established purity). The results allowed one to compare the qualitative and quantitative composition of the alkaloids in each fraction, to follow their fragmentation pathways and to classify them into four well-known *Lycopodium* types. For *L. annotinum,* almost pure alkaloid fractions were obtained using sorbents Al_2_O_3_ and silica gel with a ratio of 3:1. Fractions 1–6 collected from the experiment were dominated by alkaloids present in high percentages. On the other hand, for *L. clavatum*, the best fraction purification and the highest alkaloid recovery values were obtained using modified silanized silica gel and a reversed mobile phase system. The great advantage of this sorbent turned out to be that it allowed one to choose different solvent systems with different polarities. However, this sorbent made better use of aqueous solvent systems to separate extremely non-polar substances and to analyze specific polar substances. Nevertheless, the experiments carried out using sorbents Al_2_O_3_ and silica gel in different ratios proved to be efficient and allowed one to obtain known AChE inhibitors such as HupA and HupB. However, the predominant compounds were mainly lucidine B derivatives, but also new ones not yet identified. Moreover, using TLC bioautography in accordance with Mroczek, one observed the presence of strong AChE inhibitors for this species, as well as the presence of moderate and weaker inhibition zones for two *Lycopodium* species. In addition, the entire process was carried out in a modern device that completely declassified manual reagent spraying. As a result, the total analysis time was reduced, and the consumption of the expensive enzyme decreased twofold. In addition, cytotoxicity assays showed that extracts of three plant species of the Lycopodiaceae family were non-toxic to skin fibroblasts, raising hopes for their safe use in treating many disorders including neurodegenerative diseases. These satisfactory results set the stage for subsequent studies, which will involve further purification of the fractions to isolate pure compounds responsible for AChE inhibitory activity.

## Figures and Tables

**Figure 1 molecules-26-06379-f001:**
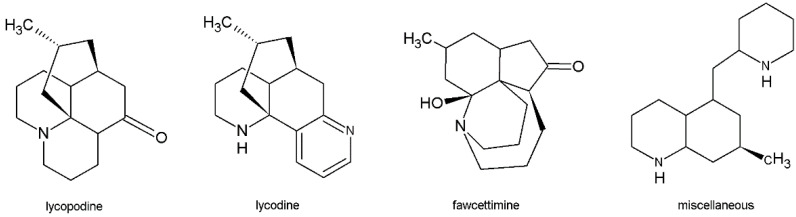
Representative compounds of the four main classes of *Lycopodium* alkaloids.

**Figure 2 molecules-26-06379-f002:**
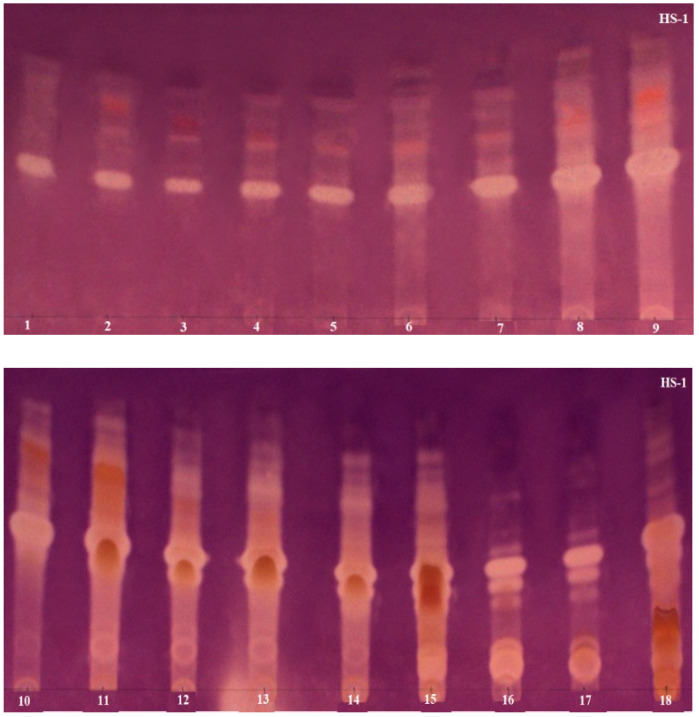
TLC chromatograms presenting *H. selago* fractions obtained from two experiments after acetylcholinesterase assays. HS-1 shows 18 fractions obtained from the first experiment via VLC using Al_2_O_3_ and silica gel in ratio 1:3 as sorbents; HS-2 shows 16 fractions obtained from the second experiment via VLC using Al_2_O_3_ and silica gel in ratio 3:1 as sorbents.

**Figure 3 molecules-26-06379-f003:**
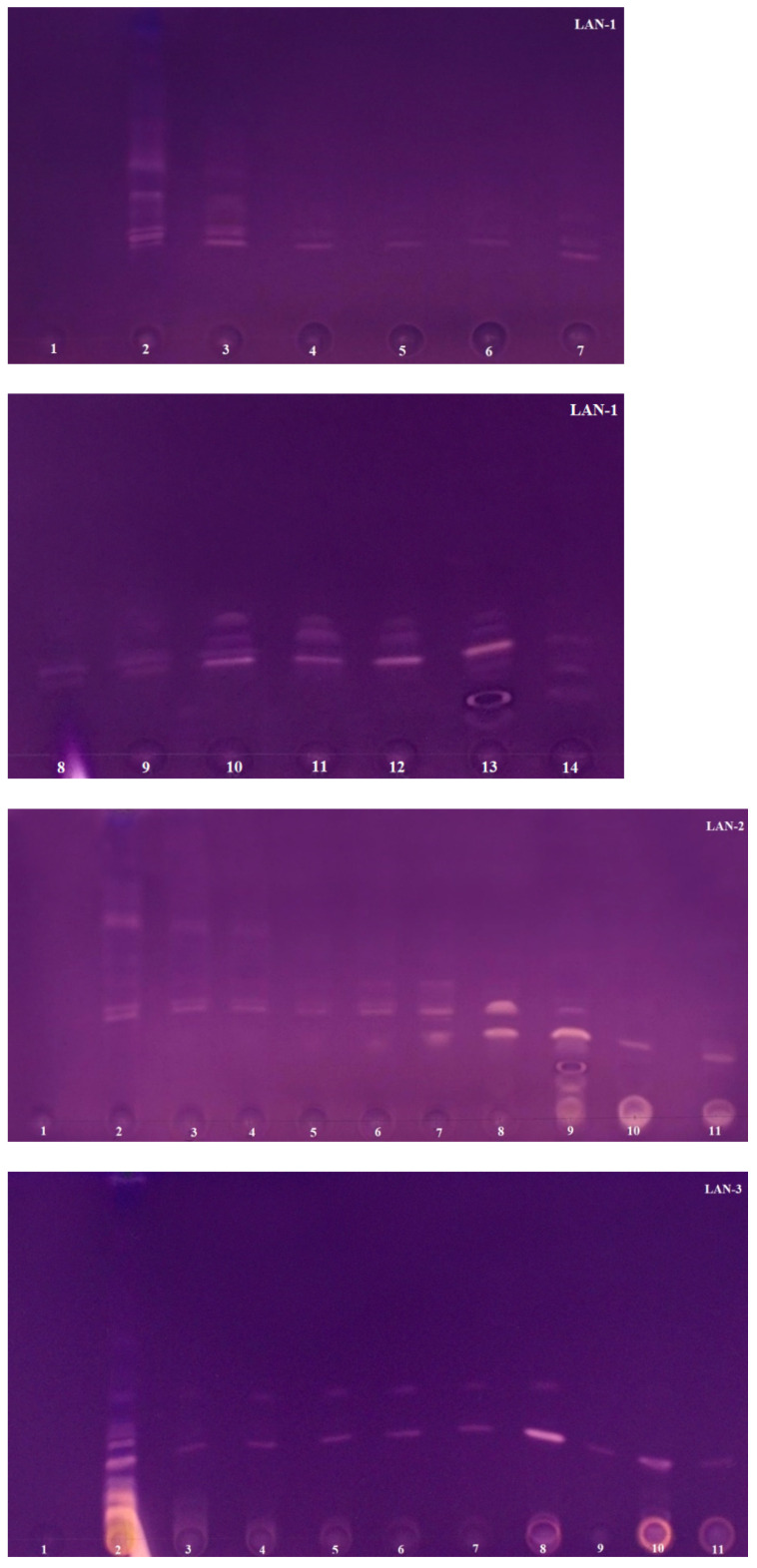
TLC chromatograms presenting *L. annotinum* fractions obtained from three experiments after acetylcholinesterase assays. LAN-1 shows 14 fractions obtained from the first experiment via VLC using Al_2_O_3_ and silica gel in ratio 1:3 as sorbents; LAN-2 shows 11 fractions obtained from the second experiment via VLC using Al_2_O_3_ and silica gel in ratio 3:1 as sorbents; LAN-3 shows 11 fractions obtained from the third experiment via VLC using silanized silica gel as sorbent.

**Figure 4 molecules-26-06379-f004:**
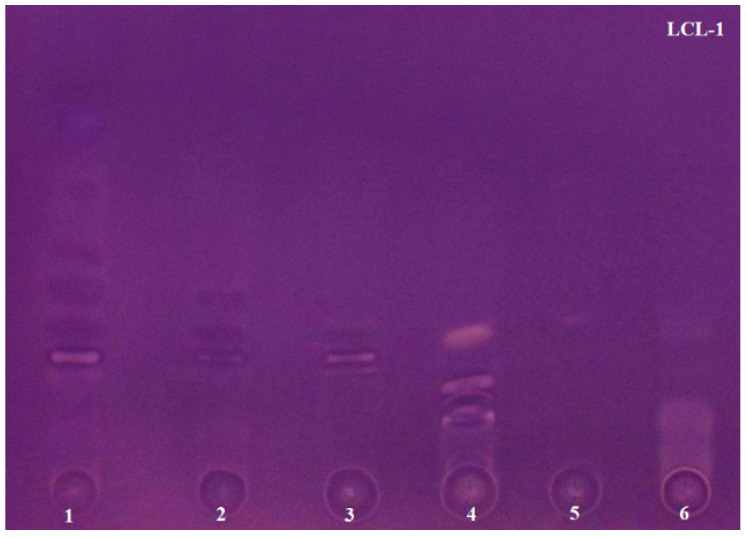
TLC chromatograms presenting *L. clavatum* fractions obtained from four experiments after acetylcholinesterase assays. LCL-1 shows 6 fractions obtained from the first experiment via VLC using Al_2_O_3_ and silica gel in ratio 1:3 as sorbents; LCL-2 shows 10 fractions obtained from the second experiment via VLC using Al_2_O_3_ and silica gel in ratio 3:1 as sorbents; LCL-3 shows 7 fractions obtained from the third experiment via VLC using silanized silica gel as a sorbent; LCL-4 shows 11 fractions obtained from the fourth experiment via VLC using silanized silica gel as a sorbent and using new reversed-phase system.

**Figure 5 molecules-26-06379-f005:**
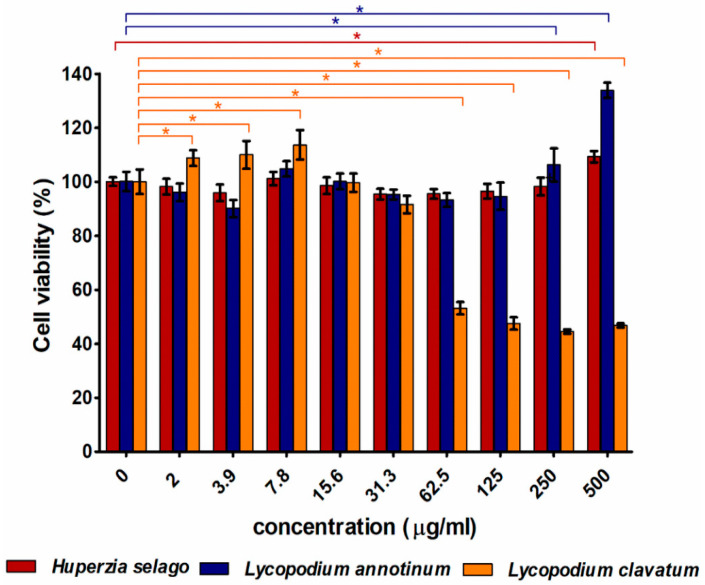
Cytotoxicity of tested extracts against human skin fibroblasts (BJ cell line) determined using MTT assay after 24 h exposure time; * *p* < 0.05—statistically different results compared to the control cells (concentration = 0 μg/mL).

**Table 1 molecules-26-06379-t001:** Fractions obtained in experiments carried out for three plant species of the Lycopodiaceae family using different combinations of sorbents and different mobile phases using the VLC method.

The Number of Experiments	1			2			3		4
Solvent gradient system used	95:5:0.2 (*v*/*v*/*v*) chloroform: methanol: 25% aqueous ammonia solution								60% aqueous methanol solution with 2 drops of 10% aqueous tartaric acid 85% aqueous methanol solution with 2 drops of 10% aqueous tartaric acid
Sorbent filling ratio	1:3 Al_2_O_3_ (17 g):silica gel (25 g)			3:1 Al_2_O_3_ (57 g):silica gel (8 g)			silica gel 60 H silanized (26 g)		silica gel 60 H silanized (26 g)
Plant materials	LCL	LAN	HS	LCL	LAN	HS	LCL	LAN	LCL
Number of fractions obtained	6	14	18	10	11	16	7	11	11

LCL—*Lycopodium clavatum* L., LAN—*Lycopodium annotinum* L., HS—*Huperzia selago* L.

**Table 2 molecules-26-06379-t002:** Results of the alkaloids identified via HPLC/ESI-QTOF-MS present in fractions 1-14 obtained from experiment 1 via VLC using sorbents Al_2_O_3_ and silica gel in ratio 1:3 from *L. annotinum* extract.

Isolated Compounds	Fraction Number													
	1	2	3	4	5	6	7	8	9	10	11	12	13	14
Acrifoline	-	-	-	-	-	4.5	-	-	12.6	16.0	20.7	22.7	21.6	6.4
Lycopodine	-	-	-	-	-	22.1	28.6	22.9	16.8	23.5	17.3	21.0	8.1	-
Annotinine	-	85.7	52.6	9.8	-	-	-	-	-	-	-	-	-	-
Dihydroannotine	-	-	-	-	-	-	-	-	-	15.0	26.0	29.1	12.5	-
Lycodine	-	1.6	14.6	28.5	32.2	25.9	28.4	7.0	3.1	0.7	-	-	-	-
Lyconnotine	-	-	-	6.1	7.8	-	-	-	-	-	-	-	41.9	-
Annotine *N*-oxide	-	-	16.8	41.3	14.3	14.6	12.4	-	-	-	-	-	-	22.3
Huperzinine	-	-	-	2.1	1.7	1.6	1.8	-	-	-	-	-	-	-
Lyconadine A	-	-	-	2.3	1.9	2.5	2.8	1.2	0.9	-	-	-	-	-
Lycopodine *N*-oxide	-	-	-	-	-			59.8	56.8	38.5	31.9	23.5		3.1
Acetylfawcettiine	-	-	-	-	-	-	-	-	-	-	-	-	-	4.8
8-β,11-α Aihydroxylycopodine	-	-	-	-	-	-	-	-	-	-	-	-	-	43.4
Des-*N*-methyl-α-obscurine	-	-	-	-	-	-	-	-	-	-	-	-	4.5	5.5
% SUM	-	87.3	84.0	90.2	57.9	71.1	73.8	90.9	90.3	93.7	95.8	96.2	88.5	85.4

**Table 3 molecules-26-06379-t003:** Results of alkaloids identified via HPLC/ESI-QTOF-MS present in fractions 1–11 obtained from experiment 2 via VLC using sorbents Al_2_O_3_ and silica gel in ratio 3:1 from *L. annotinum* extract.

Isolated Compounds	Fraction Number										
	1	2	3	4	5	6	7	8	9	10	11
Acrifoline	-	-	1.9	-	-	2.0	17.4	25.1	18.9	14.5	3.2
Lycopodine	-	-	14.5	-	12.5	73.9	41.1	12.0	-	-	-
Annotinine	85.2	89.0	71.3	87.9	61.7	-	-	-	-	9.2	5.6
Fawcettimine	-	-	-	-	-	-	-	7.2	17.5	8.3	-
Annotine	-	-	-	-	-	-	-	14.4	-	-	-
Dihydroannotine	-	-	-	-	-	-	-	8.1	16.6	-	4.5
Dihydroannotinol	-	-	-	-	-	-	-	6.5	11.6	-	4.7
*N*-Methyl dihydrolycodine	-	-	-	-	-	-	-	6.9	6.9	4.7	8.0
Lycodine	1.4	1.4	3.4	1.2	3.6	5.8	3.5	-	-	-	-
Dihydrolycopodine	-	-	-	-	-	-	-	-	-	18.1	7.0
Deacetyllycofawcine	-	-	-	-	-	-	-	-	-	-	3.2
Lycopodine *N*-oxide	-	-	-	-	4.6	5.8	28.0	9.1	18.4	-	-
Deacetylfawcettiine	-	-	-	-	-	-	-	-	-	9.2	26.3
α-Obscurine	-	-	-	-	-	-	-	-	1.5	4.1	2.2
Lyconnotine		8.2	2.6	3.8	6.2	4.7	6.1	8.4	7.2	7.9	-
Unidentified 474.3794 C_26_H_52_NO_6_	-	-	-	-	-	-	-	-	-	22.7	-
Unidentified 459.2649 C_29_H_35_N_2_O_3_	-	-	4.7	6.1	6.2	6.4	2.1	-	-	-	-
% SUM	86.6	98.6	98.4	99.0	94.9	98.5	98.1	97.6	98.5	98.8	64.6

**Table 4 molecules-26-06379-t004:** Results of alkaloids identified via HPLC/ESI-QTOF-MS present in fractions 1–11 obtained from experiment 3 via VLC using silanized silica gel as a sorbent from *L. annotinum* extract.

Isolated Compounds	Fraction Number										
	1	2	3	4	5	6	7	8	9	10	11
Acrifoline	-	16.2	17.8	21.6	28.2	23.5	28.3	63.6	30.4	-	12.4
Lycopodine	0.9	15.8	-	-	-	-	-	-	-	-	-
Annotinine	-	27.4	43.4	34.3	30.2	36.6	31.1	-	-	-	-
Annotine	-	-	-	-	-	1.99?	1.9	-	-	-	-
Dihydroannotine	-	-	-	-	-	-	-	-	7.6	18.8	11.8
Fawcettimine	-	9.9	8.7	13.7	11.4	13.5	15.8	13.7	34.1	11.5	6.7
β-Obscurine	-	-	1.4	2.3	1.1	1.3	1.7	1.2	-	-	2.8
Lycopodine *N-*oxide	-	-	-	-	-	-	-	4.1	7.4	9.2	-
Lycodine	-	3.0	1.5	1.6	-	-	-	-	-	-	-
Deacetylfawcettiine	14.0	-	-	-	-	-	-	5.3	1.8	14.9	4.5
Lyconnotine	-	15.8	19.3	18.5	21.0	16.1	11.6	3.2	-	-	-
Huperzinine	-	-	-	-	-	-	-	-	2.5	3.9	7.4
*N*-Methyl lycodine	-	-	-	-	-	-	-	-	-	5.5	14.1
Flabelline	-	3.5	2.3	4.2	2.1	2.0	1.5	-	-	-	-
Unidentified 474.3794 C_26_H_52_NO_6_	12.6	-	-	-	-	-	-	4.6	11.8	14.2	-
% SUM	27.4	91.7	94.4	96.2	94.0	93.0	91.9	95.7	95.5	78.1	59.7

**Table 5 molecules-26-06379-t005:** Results of alkaloids identified via HPLC/ESI-QTOF-MS present in fractions 1–18 obtained from experiment 1 via VLC using sorbents Al_2_O_3_ and silica gel in ratio 1:3 from *H. selago* extract.

Isolated Compounds	Fraction Number																	
	1	2	3	4	5	6	7	8	9	10	11	12	13	14	15	16	17	18
Selagoline	-	-	-	-	-	-	-	-	-	6.1	18.8	14.7	26.7	29.1	22.9	4.1		18.1
Lycopodine *N*-oxide	-	-	-	-	-	-	-	-	-	12.0	24.7	38.7	27.0	26.4	22.0	4.1	20.6	35.6
Anhydrolycodoline	1.1	1.2	1.0	0.9	-	-	-	-	-	-	-	-	-	-	-	-	-	-
8-β,11-α Dihydroxy-lycopodine	-	-	-	-	-	-	-	-	-	-	-	-	-	-	4.2	18.0	30.2	
Lycoposerramine B	-	-	2.2	1.8	1.2	1.9	1.8	1.7	3.2	1.2	-	-	-	-	-	-	-	-
Huperzine A	-	-	-	-	-	-	-	-	-	-	5.2	5.6	4.7	2.3	1.7	-	-	-
16-Hydroxyhuperzine B	-	-	-	-	-	-	-	-	-	-	-	-	-	-	0.8	2.2	1.3	-
Huperzine B	-	-	-	-	-	-	-	-	-	-	-	-	-	-	4.5	-	-	-
Lycodine	-	-	-	-	3.1	5.3	5.6	4.8	4.2	4.1	2.2	-	-	-	-	-	-	-
Des-*N*-methyl- β-obscurine	-	-	-	-	-	-	-	-	-	-	-	-	-	-	-	5.4	4.6	-
Dehydrooxolucidine B		19.8	51.4	33.5	57.6	55.8	49.6	54.2	45.2	14.4	3.2	6.2	5.5	3.2	4.4	-	-	-
Serratidine	2.0	4.6	4.3	3.9	2.1	0.9	1.1	1.3	1.2	-	-	-	-	-	-	1.3	1.5	-
Lucidine B	3.3	28.8	16.6	21.6	15.2	17.2	14.3	18.1	23.2	39.2	31.2	24.2	22.6	31.8	11.6	3.3		3.1
Dehydrolucidine B	-	-	3.1	3.8	4.7	3.6	4.5	5.4	5.7	4.7	2.5	3.2	2.5	4.2	14.3	41.7	24.5	5.6
Oxolucidine B	61.2	18.6	12.3	11.7	9.7	11.5	8.7	6.6	7.1	3.7	2.1	3.1	2.3	1.9	-	-	-	-
Unidentified 440.3637 C_28_H_46_N_3_O	-	-	-	-	-	-	-	-	-	-	-	-	-	-	5.7	10.8	2.8	17.9
Unidentified 456.3588 C_28_H_46_N_3_O_2_	-	-	-	-	-	-	-	-	-	-	-	-	-	-	1.2	2.9	3.3	-
SUM %	67.5	72.9	90.8	77.1	93.5	96.1	85.6	91.9	89.7	85.2	89.8	95.6	91.2	98.7	93.1	93.7	88.8	80.3

**Table 6 molecules-26-06379-t006:** Results of alkaloids identified via HPLC/ESI-QTOF-MS present in fractions 1–16 obtained from experiment 2 via VLC using sorbents Al_2_O_3_ and silica gel in ratio 3:1 from *H. selago* extract.

Isolated Compounds	Fraction Number															
	1	2	3	4	5	6	7	8	9	10	11	12	13	14	15	16
Luciduline	3.5	-	-	-	-	-	-	-	-	-	-	-	-	-	-	-
Serratidine	2.2	-	-	-	-	-	-	-	-	-	-	-	-	-	-	-
Selagoline	-	2.9	7.3	19.0	26.4	23.5	22.6	14.8	22.4	23.2	27.2	18.2	8.0	1.2	-	-
Lycopodine *N*-oxide	-	9.5	43.6	26.8	30.5	25.7	24.1	23.8	24.1	26.8	24.1	15.1	28.8	41.6	10.9	-
Dehydrooxolucidine B	67.9	50.1	18.1	25.7	11.2	18.2	19.6	20.3	29.2	12.2	6.2	9.4	-	-	21.1	43.1
Lucidine B	6.6	30.1	24.0	24.2	22.6	20.2	23.1	27.1	17.2	29.2	32.3	33.2	13.6	-	-	1.1
Dehydrolucidine B	-	-	-	-	1.9	4.8	3.2	7.0	4.1	1.9	1.2	1.6	25.2	15.2	4.1	1.4
Oxolucidine B	17.7	4.1	3.0	2.7	2.6	2.2	2.1	1.1	1.4	2.0	0.2	1.1	0.8	-	-	-
Lycodine		0.9	0.6	0.7	0.5	0.8	0.4	0.9	0.4	0.6	-	-	-	-	-	-
Lycoposerramine B	1.1	1.1	1.1	-	0.9	1.1	-	-	-	-	-	-	-	-	-	-
Fawcettimine	-	-	-	-	-	-	-	-	-	-	-	-	-	8.3	-	-
Dihydrolycopodine	-	-	-	-	-	-	-	-	-	-	-	-	3.2	1.5	15.2	14.2
Deacetyllycoclavine	-	-	-	-	-	-	-	-	-	-	-	-	-	-	3.1	7.1
8-β,11-α Dihydroxy-lycopodine	-	-	-	-	-	0.8	0.9	0.6	0.9	1.0	1.0	2.2	5.2	4.2	5.3	4.2
Huperzine A	-	-	-	-	-	-	-	-	-	-	-	8.1	6.6	0.9		
Unidentified 474.3796 C_27_H_48_N_5_O_2_ or C_26_H_52_NO_6_	-	-	-	-	-	-	-	-	-	-	-	-	-	-	8.2	10.1
Unidentified 426.3843 C_28_H_48_N_3_ or C_30_H_46_O_2_	-	-	-	-	-	-	-	-	-	-	-	-	-	6.5	-	-
Unidentified 440.3637 C_28_H_46_N_3_O	-	-	-	-	-	-	-	-	-	-	-	-	6.1	9.9	2.3	3.8
Unidentified 456.3588 C_28_H_46_N_3_O_2_	-	-	-	-	-	-	-	-	-	-	-	-	2.4	1.4	6.1	-
Unidentified 438.3481 C_28_H_44_N_3_O	-	-	-	-	-	-	-	-	-	-	-	-	-	4.1	4.3	-
SUM %	98.9	98.7	97.6	99.0	96.4	97.2	96.1	95.4	99.7	96.8	92.2	88.8	99.7	94.8	80.7	84.8

**Table 7 molecules-26-06379-t007:** Results of alkaloids identified via HPLC/ESI-QTOF-MS present in fractions 1–6 obtained from experiment 1 via VLC using sorbents Al_2_O_3_ and silica gel in ratio 1:3 from *L. clavatum* extract.

Isolated Compounds	Fraction Number					
	1	2	3	4	5	6
Acetyllycofawcine	-	-	-	10.0	13.9	-
Acetylfawcettiine	-	-	-	12.2	10.9	-
Lycopodine *N*-oxide	-	-	9.1	15.5	13.3	20.6
Lycopodine	-	-	7.4	7.5	8.1	-
Flabelline	-	-	-	3.9	2.6	-
Fawcettimine	-	-	-	1.5	0.8	5.7
Deacetyldidehydrolycofawcine	-	-	-	-	6.9	-
Lycodine	-	3.9	3.9	1.4	-	-
4α,8β,12β-Trihydroxylycopodine	-	-	4.3	-	-	-
4α,6α-Dihydroxyanhydrolycodoline	-	-	-	1.3	-	-
4,6α-Dihydroxylycopodine or epimer	-	-	-	-	5.3	-
Lycoposerramine K	12.3	0.8	-	-	-	-
Lycofawcine	-	-	-	-	-	8.5
α-Lofoline or epimer	-	-	-	-	0.7	5.4
Deacetylfawcettiine	-	-	-	-	-	2.3
8β-Hydroxylycoposerramine K	-	-	-	-	-	5.7
16-Oxolyclanitin	-	-	-	5.9	-	0.3
Lycoclavanin or epimer	-	-	4.9	7.6	-	-
Serratezomine E	-	-	-	-	-	4.0
Huperzinine	0.4	2.8	3.2	-	-	-
Unidentified 371.2290 C_17_H_31_N_4_O_5_ or C_19_H_33_NO_6_	12.0	12.6	11.1	5.5	7.2	1.4
Unidentified 415.2532 C_19_H_35_N_4_O_6_	12.5	13.2	11.7	6.6	8.4	4.9
Unidentified 679.5133 C_40_H_71_O_8_ or C_41_H_67_N_4_O_4_	9.0	8.3	6.9	4.5	1.1	5.7
SUM %	46.2	41.5	62.6	83.4	79.0	64.4

**Table 8 molecules-26-06379-t008:** Results of alkaloids identified via HPLC/ESI-QTOF-MS present in fractions 1–10 obtained from experiment 2 via VLC using sorbents Al_2_O_3_ and silica gel in ratio 3:1 from *L. clavatum* extract.

Isolated Compounds	Fraction Number									
	1	2	3	4	5	6	7	8	9	10
Acetyllycofawcine	7.08	-	-	4.39	4.4	3.29	2.48	4.97	3.17	3.19
Acetylfawcettiine	10.31	6.98	4.89	7.03	7.53	7.55	6.17	7.26	3.45	2.92
Lycopodine *N-*oxide	5.57	7.03	7.28	5.16	7.46	5.26	6.71	5.03	6.88	6.24
Lycopodine	-	2.99	2.37	3.06	3.3	5.01	8.75	6.99	6.67	4.64
Flabellidine	0.78	-	-	-	-	-	1.13	1.01	2.14	2.03
Fawcettimine	-	-	-	-	-	-	-	-	1.15	6.25
Lycodine	3.78	1.61	1.38	3.67	1.46	1.96	1.83	2.96	1.91	0.93
Lycoposerramine K	3.88	1.95	0.69	0.78	0.75	0.55	-	-	-	-
α-Lofoline or epimer	-	-	-	-	-	-	-	-	1.1	3.3
Serratezomine E									4.7	6.59
Deacetylfawcettiine	-	-	-	-	-	-	-	-	-	2.37
Lycofawcine									1.12	0.89
Huperzinine	2.73	0.45	0.55	2.08	2.09	2.81	2.15	2.62	-	-
16-Oxolyclanitin									0.77	1.24
Lycoclavanin or epimer	3.31	2.11	3.12	3.75	2.41	3.15	2.13	0.88		
Unidentified 371.2290 C_17_H_31_N_4_O_5_ or C_19_H_33_NO_6_	9.74	12.05	12.1	9.48	9.48	8.06	8.55	8.07	8.29	7.09
Unidentified 415.2532 C_19_H_35_N_4_O_6_	10.51	12.77	12.67	9.63	9.8	8.68	8.87	8.23	7.67	7.57
Unidentified 679.5133 C_40_H_71_O_8_ or C_41_H_67_N_4_O_4_	6.46	7.89	7.47	8.01	8.48	6.94	6.24	5.93	4.94	4.6
SUM %	64.15	55.83	52.52	57.04	57.16	53.26	55.01	53.95	53.96	59.85

**Table 9 molecules-26-06379-t009:** Results of alkaloids identified via HPLC/ESI-QTOF-MS present in fractions 1–7 obtained from experiment 3 via VLC using silanized silica gel as a sorbent from *L. clavatum* extract.

Isolated Compounds	Fraction Number						
	1	2	3	4	5	6	7
Acetylfawcettiine	12.1	-	-	-	-	1.2	4.9
Lycopodine *N*-oxide	12.3	8.4	6.7	3.7	2.2	-	-
Lycopodine	9.7	7.3	6.5	6.2	2.2	4.3	12.6
Dihydrolycopodine	4.3	4.6	5.3	6.4	5.0	5.1	7.3
Flabelline	-	0.8	1.4	-	-	-	-
Fawcettimine	-	1.8	3.6	2.6	2.7	2.2	1.5
8β-Hydroxylycoposerramine K	-	2.4	3.6	2.8	1.5	1.9	-
Lycoposerramine K	2.9	-	-	-	-	-	-
4,6α-Dihydroxylycopodine or epimer	-	-	-	-	-	1.1	1.4
α-Lofoline or epimer	2.1	3.1	2.4	2.3	1.1	-	1.8
Deacetylfawcettiine	-	-	1.8	2.0	3.7	4.7	2.5
Serratezomine E	-	8.1	6.2	-	2.3	-	-
Japonicumin B or lycoclavanin	0.3	4.2	4.1	-	4.2	-	2.6
16-Oxolyclanitin		1.2	5.3	1.9	-	-	2.1
Unidentified 474.3794 C_27_H_48_N_5_O_2_ or C_26_H_52_NO_6_	-	-	-	-	-	-	7.2
Unidentified 371.2290 C_17_H_31_N_4_O_5_ or C_19_H_33_NO_6_	9.9	8.0	9.0	10.6	4.0	9.8	-
Unidentified 415.2532 C_19_H_35_N_4_O_6_	10.5	8.4	9.6	11.3	4.2	10.1	5.5
Unidentified 679.5133 C_40_H_71_O_8_ or C_41_H_67_N_4_O_4_	7.39	6.7	6.87	7.12	4.75	8.17	3.91
% SUM	71.43	64.95	72.29	56.76	37.64	48.59	53.24

**Table 10 molecules-26-06379-t010:** Results of alkaloids identified via HPLC/ESI-QTOF-MS present in fractions 1–11 obtained from experiment 4 via VLC using silanized silica gel as a sorbent and reverse phase system from *L. clavatum* extract.

Isolated Compounds	Fraction Number										
	1	2	3	4	5	6	7	8	9	10	11
Lycopodine	0.8	-	28.5	31.8	27.6	30.0	33.5	39.1	36.8	35.4	37.2
Dihydrolycopodine	-	-	8.2	12.3	12.7	8.5	6.2	6.0	7.6	11.3	7.0
Lycopodine *N*-oxide	-	-	-	8.5	11.8	13.0	10.1	8.8	11.0	13.2	13.2
8β-Hydroxylycoposerramine K	-	46.3	22.8	13.3	8.9	7.3	3.7	4.8	3.2	5.5	4.3
Acetylfawcettiine	-	-	-	-	-	-	1.1	0.9	-	0.8	1.1
8-β,11-α Dihydroxylycopodine or epimer	-	-	3.9	-	-	1.0	-	1.2	0.9	0.7	1.0
Lycodine	-	-	-	1.2	1.1	0.8	-	-	1.1	0.9	0.6
α-Lofoline or epimer	-	-	-	1.1	1.1	1.0	1.1	1.1	2.0	1.0	1.2
β-Obscurine	-	-	-	1.2	2.1	1.2	-	-	-	-	1.1
Deacetylfawcettiine	-	1.1	4.6	3.2	3.5	4.6	2.2	1.7	3.6	1.9	3.4
Serratezomine E	-	-	-	1.5	2.9	1.5	1.0	1.3	1.2	3.2	3.6
*N*-Methyl lycodine	-	-	-	-	-	-	-	-	0.5	0.6	0.8
Unidentified 236.1489 C_10_H_22_NO_5_	-	20.1	11.7	10.3	9.8	10.0	9.3	9.9	11.1	12.5	11.5
SUM %	0.8	67.5	79.8	84.4	81.4	78.7	68.0	74.7	79.0	87.0	85.8

## Data Availability

The data confirming the results obtained were deposited in the Independent Laboratory of Chemistry of Natural Products and Independent Unit of Tissue Engineering and Regenerative Medicine, Chair of Biomedical Sciences, Medical University of Lublin. They are available from the corresponding author upon reasonable request.

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
