# Peer review of "Fractionation of Lycopodiaceae Alkaloids and Evaluation of Their Anticholinesterase and Cytotoxic Activities"

_molecules, 2021, doi:10.3390/molecules26216379_

Round 1

Reviewer 1 Report

Authors spent great efforts to study the Lycopodiaceae alkaloids in three plant species Lycopodium clavatum L. Lycopodium annotinum L. and Huperizia selago (L.). PLE was used for the extraction. Crude extracts were fractionated using VLC with different sorbents and eluting solvent system.  Alkaloids in all the fractions were identified by HPLC/ESI- QTOF MS by looking at both MS and MS2 data. Subsequent AChE inhibition activities were studies using TLC bioautography. Cytotoxicity was accessed using human skin fibroblasts. A few comments are listed as below.

  1. The manuscript is a bit lengthy especially the introduction and conclusion. Authors may consider making them more concise. Author can consider breaking long sentences into a few shorter sentences for easy understanding. Please also consider using simple words. For example, in line 54, “in the early second half” can be replaced with “in the middle”. In line 175, “more non-polar” can be replaced with “less polar”.
  2. Authors may not need to propose so many fragmentation pathways. One proposed pathway for each type of alkaloids will be sufficient.
  3. Alkaloid fractionation and isolation procedures can be converted into flow chart for easier understanding and comparison.
  4. Table 2-10 can be put into supplementary data.
  5. In line 81 to 82, it stated that the aim was to search for alternative sources…. While in line 147, it stated there are endangered species which may not be suitable to be alternative source. Please clarify.
  6. I noticed that there were 4 extracts of Lycopodium clavatum, 3 extracts of Lycopodium annotinum, and 2 extracts of Huperizia selago. Each extract subjected to different fractionating procedure. What are the reasons to have different number of extracts for the three species?
  7. Line 180, improving chromatographic separation.
  8. Line 206, based on hydrophilic interaction.
  9. Line 216, compound molecular mass..
  10. Line 219 to 220, …because their m/z values of protonated molecular ions
  11. Line 357, revise phase chromatographic system with higher polarity of mobile phase…
  12. Line 525, …LCL3 shows…
  13. Line 565, sample preparation

Author Response

-

Reviewer 2 Report

Materials section: How was the material dried, in the oven, in the air or is it lyophilized? Is the water content determined?

Method section: Was the analytical silica column heated during the analysis? If yes, add temperature

The discussion is written briefly and I do not make many comments. However, I recommend re-editing of TLC figures (Figures1-3) they should be resized or moved to a supplementary section.

The conclusion section is too long. It should be rewritten with clear and concise results.

The following points raised in the manuscripts should be addressed before this manuscript could be accepted.

Reviewer 3 Report

 This study aimed to analyze the alkaloid compounds and search for the effective AChE inhibitors from three  known Lycopodium species. For the achievement of this purpose the isolation of alkaloids was carried out using optimized PLE and the extracts were purified by innovative VLC with different adsorbent systems in different ratios and various mobile phase systems . This work allowed the quick comparison between the results from each experiment for the three studied species, and obtained individual compounds in each fraction. All of the fractions were thoroughly analyzed by HPLC coupled with a mass spectrometer (ESI-QTOF-MS) and assessed for AChE inhibition activity by a TLC bioautography method which enabled the detection of more than 50 compounds with several new ones.  An in vitro cytotoxicity test against human skin fibroblasts  performed for the three plant extracts further revealed they are not cytotoxic toward this kind of cells. Overall, this study led to the useful results for the discovery of effective fractions and compounds with potent AChe inhibition activity from three plants investigated and was recommended to be accepted for publication in Molecules, with the clarification or improvement of the several points listed below.

1. It was claimed in the abstract that several new compounds are previously unreported. What are the new compounds discovered in this study?

2. In  L55-57, it was described that more than 200 alkaloids from Lycopodium species, which were classified by Canadian scientists Ayer and co-leaders [17] in the 1990s into four main classes: lycopodine, lycodine, fawcettimine and miscellaneous alkaloids. For the interest and convenience of readers, please give the structures of the representative compounds of each class. 

3. In Tables 2-10, it was shown that in HPLC separation several compounds were shown the presence in discrete fractions obtained by experiments from VLC. This seems to be not reasonable as the same compound should be isolated from the same detected peak of successive fractions.

4. The molecular structures of the active compounds found in this study should appear in this manuscript.

5. In Supplementary materials,  many ion fragmentation sequences were found to be incorrect. 

Reviewer 4 Report

The manuscript presents the fractionation of alkaloids obtained from three plant species of the Lycopodiaceae family by PLE and gVLC, and their analysis by HPLC-ESI-QTOF-MS for the identification of the obtained alkaloid fractions. The anti-AChE activity of alkaloid fractions was then evaluated by a TLC approach and the cytotoxicity was also studied. In overall, the manuscript is well presented, the methods are appropriately conducted, and the obtained results are interesting to the readers of Molecules journal. Moreover, the identification of novel alkaloids and the detection of unidentified substances enables a promising research field for future studies.  

Minor comments:

  • Section 2. Results and Discussion is presented before Section 3. Materials and Methods. Consider to switch the order of presentation of these sections in the manuscript.
  • Line 177 and Table 1. Replace the expression "two drops" of added 10% tartaric acid for the real volume in order to improve repeatability for other laboratories (it should be around 40 microliters probably).
  • Lines 608-609. In the analytical literature it is common to denonate the aqueous mobile phase solvent as solvent A and the organic mobile phase solvent as solvent B. In the manuscript, these solvents A and B are reversed.
  • In the reviewer's opinion, the numer of Tables (10 tables) and Figures (4 figures) in the manuscript is at some point excessive and it could be repetitive for the broad audience of the journal. Moreover, 100 figures are included in the Supplementary materials. Please, consider to reduce the number of Tables and Figures, maybe merging some of them or summarizing the most important results for a clearly presentation. 

According to the above comments and suggestions, the reviewer's overall recommendation is to Accept the manuscript after a minor revision for its publication in Molecules journal.

Round 2

Reviewer 2 Report

The author accepted all suggested comments. The manuscript can be published in present form.

This manuscript is a resubmission of an earlier submission. The following is a list of the peer review reports and author responses from that submission.